# Effect of Parity, Body Condition Score at Calving, and Milk Yield on the Metabolic Profile of Gyr Cows in the Transition Period

**DOI:** 10.3390/ani13152509

**Published:** 2023-08-03

**Authors:** José Carlos dos Santos Breda, Elias Jorge Facury Filho, Karina Keller da Costa Flaiban, Julio Augusto Naylor Lisboa

**Affiliations:** 1Department of Veterinary Clinic, Veterinary School, Universidade Estadual de Londrina, Londrina 86057-970, PR, Brazil; 2Department of Veterinary Clinic and Surgery, Veterinary School, Universidade Federal de Minas Gerais, Campus Pampulha, Belo Horizonte 31270-901, MG, Brazil; facury@vetufmg.br

**Keywords:** dairy cow, transition period, metabolic disorders, Zebu

## Abstract

**Simple Summary:**

Zebu cattle can produce food in high quantity and quality under adverse farming conditions due to their greater tolerance to heat stress and resilience to parasites compared to taurine breeds. With advances in genetic improvement programs, Gyr dairy cows currently have production rates comparable to those of some taurine herds. The present study aimed to characterize the metabolic profile (energy, protein, and mineral metabolism) of Gyr cows during the transition period, which comprises the 21 days before and after parturition. This period constitutes a fundamental phase in the life of a dairy cow, as significant hormonal and metabolic changes occur, which can predispose the cow to diseases, in addition to affecting its productive indices. This study provides relevant information by showing that high-yielding Gyr cows are metabolically balanced and less predisposed to disease during the transition period compared to taurine dairy breeds. The possibility of rearing this Zebu breed in the tropical regions of the world with excellent production rates highlights the importance of this subject.

**Abstract:**

This study aimed to evaluate the effects of parity, body condition score (BCS) at calving, and milk yield on the metabolic profile of Gyr (Zebu) cows. Healthy cows in late pregnancy were grouped according to parity (primiparous, biparous, and multiparous); to BCS scale at calving (high—HBCS and normal—NBCS); and to milk yield (high—HP and moderate—MP production). BCS was assessed, and blood samples were collected on −21, −7, 0, 7, 21, and 42 days relative to parturition. The concentrations of non-esterified fatty acids (NEFA), beta-hydroxybutyrate (BHB), cholesterol, glucose, total protein (TP), albumin, total calcium (Ca), phosphorus (P), and magnesium (Mg); and activities of aspartate aminotransferase and gamma-glutamyltransferase were measured. Data were analyzed by two-way repeated measures ANOVA. The frequencies of high lipomobilization, subclinical ketosis, subclinical hypocalcemia (SCH), and the occurrence of diseases during early lactation were established. Regardless of grouping, NEFA, BHB, and cholesterol increased during early lactation; glucose showed higher values at calving; TP and albumin were higher at 21 and 42 DIM; and Ca, P, and Mg were lower at calving. Parity had little effect on the metabolic profile, HBCS did not differ from NBCS cows, and HP did not differ from MP cows in most metabolites. High lipomobilization in early lactation and SCH at calving were the most common imbalances but were not related to postpartum diseases. High-yielding Gyr cows have a balanced metabolic profile during the transition period, with few biologically relevant effects of parity, BCS at parturition, or milk yielded.

## 1. Introduction

Gyr cattle (*Bos taurus indicus*) are characterized by their ability to produce milk in pasture systems under adverse environmental conditions due to their tolerance to heat stress and parasites [1]. In Brazil, Gyr cows have been selected for milk yield since the 1930s [2]. Within specialized breeding programs, some Gyr herds have stood out for their high production rates, which can be compared with some specialized dairy herds of taurine breeds [3,4]. The importance of the Gyr breed in Brazilian dairy farming is due, in addition to milk production, to its use in crosses with the Holstein breed, which produces Girolando cattle (Holstein × Gyr) that are extremely productive and adapted to the environmental conditions typical of tropical or subtropical climates [5].

The transition period is critical for dairy cows [6,7]. The endocrine and metabolic changes that occur in the pre and postpartum periods are responsible for a series of imbalances that predispose cows to diseases such as hypocalcemia, ketosis, retained placenta, metritis, mastitis, and abomasal displacement [8,9,10]. High-producing Holstein and Jersey cows are already known to be susceptible to these diseases, especially during early lactation, and have been widely studied with regard to metabolic imbalances during the transition period [11,12,13].

Information on this subject in Zebu dairy cows is scarce. Some studies conducted by Brazilian researchers on crossbred Holstein × Gyr cows have shown that high-producing cows have metabolic imbalances similar to those present in Holstein cows [14,15,16,17]. Specifically, for Gyr cows, information on the metabolic aspects during the transition period is limited to two studies that included cows with low milk production (6 to 9 kg/day) and measured a smaller number of metabolites [18,19]. The authors are unaware of previous studies on high-yielding Gyr cows, and due to the relevance of the subject, they raised the hypothesis that metabolic behavior during the transition period may be influenced by the order of lactation, body condition score (BCS) at calving, and volume of milk produced. The present study aimed to evaluate the effects of parity, BCS at calving, and milk production levels on energy, protein, and mineral metabolism during the transition period in Gyr cows selected for a high milk yield.

## 2. Material and Methods

This was a longitudinal study with convenience sampling. This study was carried out in accordance with Brazilian law and the National Council for Animal Experiment Control (CONCEA) guidelines. The project was previously approved by the Ethics Committee on the Use of Animals (CEUA) of the Universidade Estadual de Londrina (protocol number: CEUA-UEL 9250.2018.90).

### 2.1. Animals and Management

This study was carried out between June and December 2019 in a commercial dairy farm located in the state of Minas Gerais, Brazil (20°17′29″ S 45°32′23″ W and 740 m of altitude). The rainy season (November to March) is humid and cloudy with an average monthly minimum temperature of 18 °C and maximum of 31 °C. The dry season (May to September) is generally clear, with a minimum temperature of 12 °C and a maximum of 26 °C (public data from the National Institute of Meteorology). The herd comprised 154 lactating Gyr cows with mean body weights varying between 370 kg (primiparous cows) and 520 kg (multiparous cows), and an average individual daily milk yield ranging from 19 to 33 kg. Total daily production varied between 3000 and 3700 kg of milk.

Eighty-five healthy cows at approximately 240 days of gestation were included based on parity and BCS. They were divided into primiparous (P; *n* = 26), biparous (B; *n* = 21), and multiparous (M; 3–6 calvings; *n* = 17) groups, all with BCS at calving ranging from 3.0 to 3.5, according to the traditional scale [20]. Two other groups were composed according to BCS at calving, including high BCS (HBCS; >3.5; *n* = 21) and normal BCS (NBCS; 3.0–3.5; *n* = 21). The HBCS group was composed exclusively of multiparous cows. For the NBCS group, all cows from the M group were included, in addition to 4 cows drawn from the B group. It was not possible to include cows that calved with a low BCS because this condition was not observed during the study period. Finally, most of the cows included in the study, except the P cows, were redistributed into the following two other groups according to milk yield at 60 days in milk (DIM): a high producing group (HP; >30 kg of milk; *n* = 22) and a moderate producing group (MP; 20–30 kg of milk; *n* = 30). The HP group consisted of 5 biparous cows and 17 multiparous cows, while the MP group consisted of 13 biparous cows and 17 multiparous cows. Cows with dystocia or twin pregnancies were not included in the study.

The cows were kept in an open confinement system and separated into the following two batches according to the production period: non-lactating cows in late pregnancy (the last 21 days before calving) and lactating cows. Each batch was housed in uncovered paddocks, with an available space of 10–15 m^2^ per cow and a feeding trough located along the lateral fence of the paddock. Primiparous cows were housed together with other cows in the paddocks. The feed consisted of a total mixed ration (TMR) calculated for the cows in the prepartum transition period and during lactation (Table 1). The TMR was provided three times daily (at 5 a.m., 11 a.m., and 4 p.m.) to ensure that feed was always available in the trough. Water was provided ad libitum from troughs that were easily accessible to the animals.

The calves were separated from the cows immediately after calving, and the colostrum was milked within one hour postpartum using a mechanical milking system. Throughout lactation, the cows were manually milked twice daily (at 5 a.m. and 3 p.m.). Individual milk yields were measured by means of weighing every 15 days.

### 2.2. BCS Assessment and Sample Collection and Processing

The BCS was always evaluated by the same trained examiner at −21 (±3), −7 (±2), 0, 7, 21, and 42 days relative to calving, using the traditional scale [20].

At the same time points, blood samples were collected by coccygeal venipuncture using a 30 × 0.8 mm needle and vacuum tubes without an anticoagulant and with EDTA and sodium fluoride. On the day of calving, blood samples were collected immediately after milking the colostrum. At other time points during lactation, samples were collected after the first milking of the day. After a 60 min rest, the samples were centrifuged (2000× *g* for 10 min). The serum and plasma obtained were stored in 2 mL aliquots and frozen (−20 °C) until processing six months later.

Serum concentrations of non-esterified fatty acids (NEFA) (NEFA; Randox Laboratories Ltd., Crumlin, UK); beta hydroxybutyrate (BHB) (Rabut; Randox Laboratories Ltd., Crumlin, UK); cholesterol (Cholesterol; Flex reagent cartridge, Siemens, São Paulo, SP, Brazil); total protein (TP) (TP; Flex reagent cartridge, Siemens); albumin (ALB-PP; Gold Analisa Diagnóstica Ltd., Belo Horizonte, MG, Brazil); total calcium (Ca) (Ca; Flex reagent cartridge, Siemens); phosphorus (P) (P; Flex reagent cartridge, Siemens); and magnesium (Mg) (Mg; Flex reagent cartridge, Siemens); plasma glucose concentration (Glucose; Flex reagent cartridge, Siemens); and serum aspartate aminotransferase (AST) (AST; Flex reagent cartridge, Siemens) and gamma-glutamyltransferase (GGT) (GGT; Flex reagent cartridge, Siemens) activities were measured in an automated spectrophotometer (Dimension Xpand Plus, Siemens, São Paulo, SP, Brazil) using specific commercial reagents.

### 2.3. Definition of Diseases and Metabolic Imbalances

The cows were monitored daily, and the diseases that occurred during early lactation (up to 60 DIM) were evaluated. Uterine secretions were evaluated by Metrichek^®^ at 7, 21, and 42 DIM for the diagnosis of endometritis or metritis according to the established classification [21]. Clinical mastitis was diagnosed when visible changes were observed in the milk, with or without signs of inflammation in the affected mammary gland.

The cutoff points adopted to define high lipomobilization were as follows: prepartum NEFA concentrations >0.3 mmol/L [22] or >0.4 mmol/L [23] at two consecutive sampling times and postpartum concentrations >0.7 mmol/L at two consecutive sampling times [22,23]. Subclinical ketosis (SK) was defined by postpartum BHB concentrations >1.2 mmol/L [22]. The value of 2.5 mmol/L was defined as the cutoff point for hypoglycemia [24]. Subclinical hypocalcemia (SCH) was defined as a total Ca concentration of <2.125 mmol/L [25,26,27].

### 2.4. Statistical Analysis

The Shapiro–Wilk and Brown–Forsythe tests were used to verify the Gaussian distribution and equality of variance, respectively. Two-way repeated measure ANOVA was used to test the effects of time relative to calving (−21 × −7 × 0 × 7 × 21 × 42 days), the effect of parity (P × B × M), and the interaction between these two factors. The same analysis was applied to the cases of BCS at calving (HBCS × NBCS) and milk yield (HP × MP). All measured metabolites and BCS were included in the analyses, and the variables are presented as means and standard error or standard deviations. The milk yield at 60 DIM, average daily milk yield, total DIM, and total milk yield were compared between groups using one-way ANOVA. Tukey’s test was used for multiple comparisons. The probability of error for all the tests was 5%. The analyses were performed using SigmaPlot for Windows 13.1 (Systat Software Inc., San Jose, CA, USA).

## 3. Results

The milk yield results for all groups are presented in Table 2. Parity did not interfere with lactation duration (*p* = 0.500), but P cows had a lower milk yield at 60 DIM (*p* = 0.007) and lower average daily production throughout lactation (*p* = 0.001) than B and M cows. The total milk yielded during lactation was lower in P cows than in M cows (*p* = 0.005). Milk yield did not differ between HBCS and NBCS cows (*p* > 0.05) but was higher in HP cows than in MP cows (*p* < 0.001). On average, HP cows had 1 month longer lactation time than MP cows (*p* = 0.020).

The BCS differed between groups related to parity and BCS at calving but showed no difference when the cows were grouped according to milk yield (Table 3, Table 4 and Table 5). Regardless of how the cows were grouped, BCS remained unchanged until calving and decreased during early lactation (Table 3, Table 4 and Table 5). P cows had a lower BCS than the other groups at 42 DIM (Appendix A), and HBCS cows maintained higher values than NBCS cows at every time point, except at 42 DIM (Appendix A).

The metabolites analyzed in the serum and plasma showed variations over time relative to calving (Table 3, Table 4 and Table 5). Regardless of grouping, the NEFA and BHB values increased during early lactation, glucose showed higher values at calving, and cholesterol concentrations gradually increased until 42 DIM. The concentrations of TP and albumin were higher at 21 and 42 DIM, AST activity was higher at 7 DIM, and GGT activity was higher at calving. The concentrations of Ca, P, and Mg were lower at calving, whereas the concentration of Mg remained low at 7 DIM.

When the cows were grouped according to BCS at calving, there were no differences between the groups and no interactions between factors (Table 4). In cows grouped by parity, interactions were observed for BHB, glucose, cholesterol, GGT, Ca, and P (Table 3). P cows had lower BHB concentrations than the other cows at 42 DIM, glucose did not differ between the groups, and cholesterol was higher in B than in P cows at 21 DIM and lower in B than in M cows at 42 DIM (Appendix A). GGT activity was lower in B cows than in P cows at calving (Appendix A), and P concentrations were higher in P cows at 7 and 21 DIM (Appendix A). At calving, the Ca concentration was higher in P cows, intermediate in B cows, and lower in M cows. P cows had intermediate Ca values at 21 DIM and lower values than those of B cows at 42 DIM (Appendix A).

HP cows did not differ from MP cows in most serum metabolites (Table 5). HP cows showed higher concentrations of glucose at calving and cholesterol at 21 and 42 DIM (Appendix A). P concentrations were lower in HP cows at calving, 7 DIM, and 21 DIM (Appendix A).

Regarding metabolic imbalances, the frequency of high lipomobilization in the prepartum period was 27.0% (23/85) and 23.5% (20/85), assuming the cutoff points of >0.3 mmol/L and >0.4 mmol/L of NEFA, respectively. During early lactation, high lipomobilization was observed in 35.3% (30/85) of cows, whereas SK was present in only 11.7% (10/85) of cows. The frequency of SCH was 32.9% (28/85) at calving and only 4.7% (4/85) of cows maintained SCH at 7 DIM. Among the hypocalcemic cows, 5 were primiparous, 5 were biparous, and 18 were multiparous.

Few diseases were observed in the studied cows up to 60 DIM, including clinical mastitis (18.8%; 16/85) and metritis (11.7%; 10/85). Retained placenta, endometritis, hypocalcemic paresis, clinical ketosis, or abomasal displacement were not observed.

## 4. Discussion

Information involving the metabolism of dairy Zebu cows is scarce in the literature, with the vast majority of studies carried out on cows of taurine breeds raised in temperate climate regions. To the authors’ knowledge, this is the first study to characterize the metabolic profile of high-yielding Gyr dairy cows. Milk production of the cows studied (20–40 kg/day), including MP cows, was higher than those previously described. Values ranging from 6 to 15 kg/day have been reported in Gyr cows [18,19,28,29,30].

Even with high production, the values found in the studied cows were still below those described for high-producing Holstein cows, ranging from 40 to 50 kg/day [31,32,33], but were similar to those found in Jersey cows, ranging from 31 to 37 kg/day [34] and in Simmental cows [35]. These values are also higher than those usually reported for crossed Holstein × Gyr cows, which range from 20 to 30 kg/day [5,14,16,17,36]. Metabolic profile studies are more suitable for high-producing dairy cows due to the greater risk of metabolic imbalances in these cows [37,38].

The parity influenced the milk production in the studied cows, being higher in M than in P cows. Lower production levels are reported in first lactation cows due to differences in tissue mobilization and nutrient partition for growth and milk production [39]. Furthermore, mammary tissue is not yet fully developed in primiparous cows, as the number of secretory cells is lower, which will be reversed with maturity [40,41].

The BCS slightly varied between the groups according to parity during the transition period, showing a difference at 7 and 42 DIM when the P cows had lower values, indicating that they lost more BCS during early lactation. This corroborates the results in Holstein cows [42,43]. Regarding variation over time, there was a reduction in the BCS at the beginning of lactation, except in M cows. Loss of BCS during early lactation is common due to the mobilization of body reserves, especially fat and protein, to meet the demands for milk production [44].

In the study presented here, BCS at calving had no effect on milk production, and HBCS cows had total and mean milk production similar to that of NBCS cows. Consistent with these results, no influence of BCS at calving on milk production was found in Holstein × Gyr cows [45]. The lack of a group of Gyr cows with low BCS at calving made it impossible to carry out a more complete analysis of the effects of BCS on the milk production in these animals. Although this was planned, no cows with low BCS at parturition were found during the study period. HBCS cows had a greater loss of body condition during early lactation than NBCS cows, which is similar to previous reports on Holstein cows [46,47].

Based on the increased NEFA and BHB concentrations in early lactation, our results corroborate those previously described in Holstein cows, demonstrating no relationship between parity and NEFA and BHB concentrations [48,49]. However, higher levels of NEFA and BHB were reported in primiparous than in multiparous Holstein cows raised in pasture systems [50]. On the other hand, lower NEFA and BHB levels were found in primiparous Holstein cows [38,51]. Such findings may be influenced by numerous factors, such as time relative to calving, BCS at calving, production system (confined vs. pasture), liver function, and breed [52]. These findings suggest that under good nutritional conditions, parity does not influence NEFA and BHB concentrations before and after calving. BHB was lower in P Gyr cows just at 42 DIM, probably due to lower tissue mobilization in primiparous cows than in multiparous cows, as previously described [53].

Differences in NEFA and BHB concentrations were observed between the HBCS and NBCS cows at 42 DIM. The HBCS cows showed higher BCS values and a greater loss of BCS after calving. Similar results were previously reported in Holstein cows; those with high BCS during the prepartum period and at calving showed a greater loss of body condition and energy imbalance during early lactation [44,54,55]. Higher levels of NEFA and BHB were also found in HP versus MP cows at 42 DIM. This can be explained by the greater mobilization of body reserves expected in high-producing cows [56]. In Holstein cows, a relationship between milk production and high levels of NEFA and BHB was not found [57]; however, it is well documented that high milk yield directly influences the occurrence of energy imbalances [58,59,60].

The observed variation in NEFA and BHB values over time in Gyr cows agrees with previous reports on Holstein cows [44,54,61] and crossbred Holstein × Gyr cows [15,16,17,62]. However, lower NEFA and higher BHB values after calving were reported in Jersey cows [63]. In low-yielding Gyr cows, changes in NEFA were not observed [18], and higher NEFA and BHB values were reported in heifers with high BCS [19]. Increased lipolysis due to a negative energy balance (NEB) elevates circulating NEFA concentrations, which are partially oxidized in the liver, causing the increased production of ketone bodies, such as BHB [23].

Elevated NEFA concentrations are indicative of pre and postpartum NEB and high lipomobilization, and are associated with several diseases that occur during this period, such as clinical and subclinical ketosis, abomasal displacement, retained placenta, and metritis [43,64]. In the studied cows, it was not possible to establish this relationship due to the occurrence of a few disorders during the study period. High lipomobilization was the most frequently observed imbalance in early lactation; however, the low prevalence of subclinical ketosis indicates that Gyr cows tend to be metabolically balanced in terms of energy metabolism. More studies with Gyr and other dairy Zebu cows, with a greater number of observations, are needed to establish the association of certain cutoff points for NEFA and BHB with the occurrence of diseases in the transition period, as the critical values assumed for Holstein cows should probably not be taken as a parameter for Zebu cattle. It is probable that the definition of specific reference intervals is necessary and not only for NEFA and BHB. In the case of Gyr cows, this is even more relevant.

The higher concentration of glucose at calving is consistent with the results of Holstein [27] and Holstein × Gyr cows [14,17,65]. This peak in glucose levels is caused by elevated blood cortisol due to calving stress [24]. In a previous study [28], Gyr cows showed higher cortisol concentrations than Holstein and Holstein × Gyr cows in response to stress due to their greater reactivity, which reinforces the results of the present study.

The elevated serum cholesterol levels found at 21 and 42 DIM were similar to those reported in Holstein [66,67], Holstein × Gyr [15,17,62,65], and low-yielding Gyr cows [18,19]. Higher cholesterol levels reflect a positive energy balance that is directly related to increased dry matter intake [66,67]. Elevated cholesterol levels may result from the increased production of lipoproteins in the liver as a means of removing triglycerides, owing to the mobilization of fat stores [58,66]. The small variations in AST and GGT activities indicate that the removal of triglycerides was carried out in a way that did not harm liver function, avoiding fatty infiltration in the liver.

The TP concentrations showed differences related to parity and were lower in P cows. Similar results have been reported in Holstein cows [48]. According to these authors, older cows tended to have higher concentrations of TP, similar to those of Gyr cows. The nutrient partitioning between growth and milk production can explain these findings [39,53], which are also justified by the considerable milk production in primiparous cows, since protein mobilization is also necessary for milk production [68]. The albumin values remained within physiological limits [69]. This was mainly due to the good nutritional conditions in which the animals were kept. Variations in albumin concentrations can occur due to protein deficiencies in the diet for a long period, a decrease in its synthesis, or an increase in catabolism in situations of energy deficit [58]. The protein concentrations in the Gyr cows during the study period were normal.

Serum Ca concentrations differed between parity groups at calving, with lower values in M cows than in P cows. SCH at calving occurred more frequently in multiparous cows than in primiparous or biparous cows. Cows with three or more parturitions are more prone to this imbalance and have greater difficulties in maintaining calcemia within the physiological range at the beginning of lactation [70]. In high-producing Gyr cows, this problem appears to be similar to that observed in other breeds. Even with good nutritional management and anionic diet intake during the prepartum transition period, approximately one third of the cows had SCH at calving. This result is similar to that described in Holstein cows, with SCH rates of 2%, 40%, and 66% in primiparous, biparous, and multiparous cows, respectively [64], and in Holstein × Gyr crossbred cows, with SCH rates of 38.46% [71]. Compared with the studied cows, low-yielding Gyr cows had higher Ca concentrations over the transition period [18], demonstrating that the volume of milk yielded can influence pre and postpartum Ca levels. In the studied cows, Ca did not differ between groups according to milk yield (Appendix A), although the MP cows yielded 20–30 kg of milk per day. The absence of differences between groups contrasts with previous reports on Holstein [72,73] and Holstein × Gyr cows [71].

Ca concentrations varied over time in all groups, especially at calving. Loss of this mineral through colostrum and milk explains the occurrence of lower levels on the day of calving [74], which can lead to imbalances early in lactation [75]. Clinical hypocalcemia was not observed in any of the cows and was a rare occurrence in the observed herd.

Serum P and Mg levels varied over time; however, few differences were observed between the groups during early lactation. According to a previous report [76], P is expected to decrease at the day of calving. The same variation was found in Holstein × Gyr cows [15,17,71], and in Jersey cows [63]. In addition to the levels of Ca, the levels of P decrease on the day of parturition, owing to the loss of large amounts of this mineral in the colostrum [74]. According to a previous report [11], Mg concentrations, on the other hand, increase at calving and at the beginning of lactation in Holstein cows, but this variation was not consistently observed in the studied cows. Mg deficiency is an important risk factor for the development of hypocalcemia after calving, as it interferes with the sensitivity of bone tissue to parathyroid hormones (PTH), and its cause is low dietary levels of this mineral [70]. In the present study, SCH was not related to hypomagnesemia, probably due to the satisfactory levels of Mg in the diet.

Two points must be considered as limitations of the present study. The absence of cows with low BCS at calving compromised a more comprehensive comparison of the effects of this factor on the studied blood metabolites, preventing comparison between groups according to BCS. The small number of cows studied prevented confirmation of the existence of a relationship between high NEFA values at the beginning of lactation or low Ca values at calving and the occurrence of diseases in early lactation. Future studies with Gyr cows may clarify these doubts.

## 5. Conclusions

Gyr cows, selected for high milk yields and maintained under good nutritional conditions, had a balanced metabolic profile during the transition period. Parity, BCS at parturition, and level of milk production had few biologically relevant effects on the metabolic profile of these animals. High lipomobilization in the postpartum period and SCH at calving were the most common imbalances found; however, they were not related to the presence of diseases during early lactation.

## Figures and Tables

**Table 1 animals-13-02509-t001:** Diets supplied for high-producing Gyr cows in the prepartum transition period and in the first 100 days of lactation.

	Prepartum *		Postpartum	
	(kg/d)	(%)	(kg/d)	(%)
Components				
Corn silage	22.90	65.64	20.88	65.58
Corn	4.00	12.56	2.85	8.95
Soybean meal	3.74	11.74	2.30	7.22
Citrus pulp	1.77	5.55	1.97	6.19
Protected fat	-	-	0.33	1.03
Mineral core	0.18	0.56	0.52	1.61
Ammonium chloride	0.66	2.10	-	-
Magnesium sulfate	0.54	1.70	-	-
Dicalcium phosphate	0.05	0.15	-	-
Lactating concentrate feed	-	-	3.00	9.42
Chemical composition				
Dry matter (%)	46.51		48.57	
Net energy (Mcal/kg)	2.21		2.36	
Starch (%)	31.22		34.55	
Crude protein (%)	12.33		14.92	
NDF (%)	34.39		34.13	
ADF (%)	18.57		19.30	
EE (%)	2.46		2.63	
Ca (%)	0.30		0.58	
P (%)	0.22		0.30	
Mg (%)	0.28		0.25	
K (%)	0.75		0.94	
Na (%)	0.03		0.22	
S (%)	0.28		0.20	
Cl (%)	0.20		0.42	
DCAD (mEq/100 g DM)	−10.0		9.41	

* Period of intake: 21 days before calving.

**Table 2 animals-13-02509-t002:** Values (mean ± SD) of milk yield at 60 days in milk (DIM), average daily production, total DIM and total milk yielded by Gyr cows grouped according to parity, body condition score at calving and milk yield.

Groups	Milk Yield at 60 DIM (kg)	Average Daily Production (kg/d)	Total DIM	Total Milk Yielded (kg)
P	20.31 ± 5.12 ^b^	19.50 ± 4.84 ^b^	255.76 ± 34.85	5025.95 ± 1540.37 ^b^
B	26.94 ± 8.26 ^a^	24.97 ± 5.18 ^a^	256.85 ± 39.19	6476.70 ± 1955.57 ^ab^
M	27.74 ± 11.16 ^a^	26.31 ± 8.63 ^a^	269.00 ± 40.39	7096.30 ± 2731.87 ^a^
HBCS	28.12 ± 11.94	25.86 ± 8.25	274.42 ± 41.70	7241.83 ± 2852.56
NBCS	27.17 ± 10.74	25.75 ± 8.10	261.40 ± 42.83	6781.47 ± 2658.28
HP	37.80 ± 6.32 ^a^	33.23 ± 4.92 ^a^	284.22 ± 35.21 ^a^	9464.53 ± 1928.04 ^a^
MP	22.25 ± 4.14 ^b^	21.63 ± 3.52 ^b^	258.06 ± 38.23 ^b^	5590.09 ± 1258.61 ^b^

^a,b^ different letters represent differences between groups (*p* < 0.05). P: primiparous (*n* = 26); B: biparous (*n* = 21); M: multiparous: (*n* = 17). HBCS: high body condition score (>3.5; *n* = 21); NBCS: normal body condition score (3.0–3.5; *n* = 20). HP: high production (>30 kg/day; *n* = 22); MP: moderate production (20–30 kg/day; *n* = 30).

**Table 3 animals-13-02509-t003:** Global means of body condition score (BCS) and metabolite concentrations in high-yielding Gyr cows distributed by groups according to parity. Effect of group (G), effect of days relative to calving (D) and interaction between the two factors (G × D).

Days Relative to Parturition							SEM *	*p* Value		
	−21	−7	0	7	21	42		G ^†^	D	G × D
BCS	3.34 ^a^	3.35 ^a^	3.35 ^a^	3.27 ^ab^	3.19 ^bc^	3.16 ^c^	0.028	0.014	<0.001	0.493
NEFA (mmol/L)	0.31 ^d^	0.38 ^cd^	0.83 ^a^	0.79 ^ab^	0.52 ^c^	0.64 ^bc^	0.048	0.326	<0.001	0.146
BHB (mmol/L)	0.40 ^d^	0.45 ^cd^	0.45 ^cd^	0.67 ^a^	0.54 ^bc^	0.58 ^ab^	0.030	0.901	<0.001	0.004
Glucose (mmol/L)	3.03 ^b^	3.01 ^b^	6.30 ^a^	3.40 ^b^	3.47 ^b^	3.29 ^b^	0.162	0.400	<0.001	<0.001
Cholesterol (mmol/L)	2.57 ^d^	2.58 ^d^	2.38 ^d^	2.94 ^c^	4.46 ^b^	5.80 ^a^	0.101	0.367	<0.001	0.001
TP (g/L)	77.42 ^b^	76.46 ^b^	76.35 ^b^	78.33 ^b^	83.70 ^a^	82.28 ^a^	0.868	<0.001	<0.001	0.196
Albumin (g/L)	32.59 ^b^	33.20 ^b^	33.72 ^b^	33.57 ^b^	36.42 ^a^	35.57 ^a^	0.520	0.033	<0.001	0.379
AST (U/L)	65.40 ^b^	69.93 ^b^	73.52 ^b^	94.00 ^a^	73.06 ^b^	69.95 ^b^	3.165	0.030	<0.001	0.428
GGT (U/L)	28.43 ^b^	31.73 ^b^	43.10 ^a^	33.25 ^b^	32.72 ^b^	30.28 ^b^	1.689	0.217	<0.001	0.009
Ca (mmol/L)	2.21 ^b^	2.25 ^ab^	2.18 ^b^	2.23 ^ab^	2.27 ^a^	2.24 ^ab^	0.016	0.035	<0.001	0.002
P (mmol/L)	2.26 ^a^	2.29 ^a^	1.78 ^b^	2.14 ^a^	2.27 ^a^	2.19 ^a^	0.044	0.015	<0.001	0.012
Mg (mmol/L)	1.08 ^a^	1.04 ^a^	0.96 ^b^	0.96 ^b^	1.04 ^a^	1.03 ^a^	0.018	0.170	<0.001	0.241

* Standard error of mean. ^†^ Groups according to parity: primiparous (*n* = 26), biparous (*n* = 21) and multiparous (*n* = 17). ^a,b,c,d^ different letters represent differences between timepoints (*p* < 0.05).

**Table 4 animals-13-02509-t004:** Global means of body condition score (BCS) and metabolite concentrations in high-yielding Gyr cows distributed by groups according to BCS at calving. Effect of group (G), effect of days relative to calving (D) and interaction between the two factors (G × D).

Days Relative to Parturition							SEM *	*p* Value		
	−21	−7	0	7	21	42		G ^†^	D	G × D
BCS	3.72 ^a^	3.70 ^a^	3.69 ^a^	3.51 ^b^	3.36 ^c^	3.32 ^c^	0.038	<0.001	<0.001	<0.001
NEFA (mmol/L)	0.32 ^b^	0.49 ^b^	0.90 ^a^	0.80 ^a^	0.52 ^b^	0.89 ^a^	0.068	0.369	<0.001	0.085
BHB (mmol/L)	0.36 ^b^	0.44 ^b^	0.49 ^b^	0.69 ^a^	0.50 ^b^	0.76 ^a^	0.043	0.243	<0.001	0.085
Glucose (mmol/L)	3.08 ^b^	3.00 ^b^	6.52 ^a^	3.38 ^b^	3.46 ^b^	3.11 ^b^	0.239	0.934	<0.001	0.932
Cholesterol (mmol/L)	2.61 ^cd^	2.57 ^cd^	2.40 ^d^	2.90 ^c^	4.71 ^b^	6.54 ^a^	0.139	0.487	<0.001	0.977
TP (g/L)	80.86 ^bc^	79.28 ^c^	78.28 ^c^	79.81 ^c^	87.39 ^a^	84.08 ^ab^	1.306	0.224	<0.001	0.811
Albumin (g/L)	32.41 ^c^	33.14 ^bc^	34.37 ^bc^	34.15 ^bc^	37.17 ^a^	35.23 ^ab^	0.666	0.495	<0.001	0.865
AST (U/L)	61.40 ^c^	67.98 ^bc^	73.61 ^b^	89.75 ^a^	70.95 ^bc^	71.67 ^bc^	3.308	0.056	<0.001	0.761
GGT (U/L)	26.49 ^c^	32.49 ^bc^	41.93 ^a^	35.22 ^abc^	36.65 ^ab^	36.51 ^ab^	2.693	0.638	<0.001	0.772
Ca (mmol/L)	2.20 ^ab^	2.24 ^a^	2.15 ^b^	2.22 ^ab^	2.26 ^a^	2.26 ^a^	0.021	0.403	<0.001	0.797
P (mmol/L)	2.24 ^ab^	2.30 ^a^	1.73 ^c^	2.03 ^b^	2.16 ^ab^	2.13 ^ab^	0.055	0.710	<0.001	0.976
Mg (mmol/L)	1.07 ^a^	1.01 ^ab^	0.97 ^bc^	0.94 ^c^	1.02 ^ab^	1.01 ^ab^	0.018	0.564	<0.001	0.996

* Standard error of mean. ^†^ Groups according to BCS at parturition: HBCS, high (>3.5; *n* = 21), NBCS, normal (3.0–3.5; *n* = 20). ^a,b,c,d^ different letters represent differences between timepoints (*p* < 0.05).

**Table 5 animals-13-02509-t005:** Global means of body condition score (BCS) and metabolite concentrations in Gyr cows distributed by groups according to milk yield. Effect of group (G), effect of days relative to parturition (D) and interaction between the two factors (G × D).

Days Relative to Parturition							SEM *	*p* Value		
	−21	−7	0	7	21	42		G ^†^	D	G × D
BCS	3.65 ^a^	3.64 ^a^	3.64 ^a^	3.48 ^b^	3.33 ^c^	3.28 ^c^	0.049	0.935	<0.001	0.609
NEFA (mmol/L)	0.30 ^c^	0.43 ^bc^	0.89 ^a^	0.89 ^a^	0.59 ^b^	0.89 ^a^	0.062	0.267	<0.001	0.112
BHB (mmol/L)	0.37 ^c^	0.43 ^bc^	0.48 ^bc^	0.70 ^ab^	0.57 ^b^	0.79 ^a^	0.038	0.033	<0.001	0.112
Glucose (mmol/L)	3.05 ^b^	2.96 ^b^	6.57 ^a^	3.30 ^b^	3.39 ^b^	3.10 ^b^	0.201	0.453	<0.001	0.041
Cholesterol (mmol/L)	2.64 ^d^	2.58 ^d^	2.47 ^d^	3.04 ^c^	4.89 ^b^	6.28 ^a^	0.112	0.014	<0.001	0.004
TP (g/L)	80.46 ^b^	79.27 ^b^	78.36 ^b^	79.31 ^b^	85.62 ^a^	84.13 ^a^	1.052	0.265	<0.001	0.215
Albumin (g/L)	32.50 ^d^	33.47 ^cd^	34.72 ^bc^	34.99 ^bc^	37.93 ^a^	36.25 ^ab^	0.594	0.658	<0.001	0.057
AST (U/L)	31.13 ^c^	35.33 ^bc^	69.87 ^b^	91.46 ^a^	70.75 ^b^	71.68 ^b^	2.538	0.486	<0.001	0.436
GGT (U/L)	26.53 ^c^	32.27 ^bc^	40.04 ^a^	31.17 ^bc^	32.77 ^b^	35.97 ^a^	1.808	0.402	<0.001	0.589
Ca (mmol/L)	2.21 ^b^	2.26 ^ab^	2.16 ^b^	2.22 ^ab^	2.28 ^a^	2.29 ^a^	0.017	0.184	<0.001	0.268
P (mmol/L)	2.27 ^a^	2.29 ^a^	1.76 ^c^	1.98 ^bc^	2.14 ^ab^	2.15 ^ab^	0.045	0.018	<0.001	<0.001
Mg (mmol/L)	1.09 ^a^	1.03 ^b^	0.96 ^c^	0.96 ^c^	1.05 ^ab^	1.04 ^ab^	0.015	0.018	<0.001	0.296

* Standard error of mean. ^†^ Groups according to milk production: HP, high (>30 kg/day; *n* = 22), MP, moderate (20–30 kg/day; *n* = 30). ^a,b,c,d^ different letters represent differences between timepoints (*p* < 0.05).

## Data Availability

The data presented in this study are available on request from the corresponding author. The data are not publicly available due to [privacy and ethical].

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
