# Peer review of "Effect of Parity, Body Condition Score at Calving, and Milk Yield on the Metabolic Profile of Gyr Cows in the Transition Period"

_animals, 2023, doi:10.3390/ani13152509_

Round 1

Reviewer 1 Report

Authors described metabolic profiles pre and post calving in Gyr cows depending on BCS, parity and milk production. 

The study sould be interesting, but, in my humble opinion, the statistics procedure is not correct and, therefore, the way they present the results is wrong. I encourage authors to keep working in this manuscript.

Some other comments:

- missing line numbers

- simple summary: taurine herds?

- simple summary: metabolic profile is not just that.

- abstract is too long

- Abstract:

1. which metabolic patrameters?

2. no nulliparous?

3. say BCS scale

4. the N in BCS and production is lower than the initial one

5. the statistical procedure is not correct. A repeated measurement analysis over time has to be implemented.

6. summarize results 

- Introduction: what is low milk production (refs 18 and 19). Explain more these studies and the differences with yours

- Material and methods:

1. convenient sample size? This is subjective... Or did you implement a statistical procedure to calculate your sample size?

2. please use the correct celsius degree symbol

3. reference of wheather

4. 85 cows is not the same than 26+21+17+21+21. Then you try to explain how you divided/shortened the cows per groups but it is not clear at all

5. does your data follow a normal? why you did not use a repeated measurements over time analysis?

table 2. avoid placing superscripts when there are no differences

table 3. follow journal guidelines. Moreover, place the superscripts in alphabetical order, please. It makes easier to follow. However, a repeated measurment analysis over time should have been performed

table 3. "by groups according to parity". where do you show the results by groups? Here you have just global means of all groups... 

table 4. the same than the previous comment. It is suposed to show differences due to BCS. But, where are the BCS groups? Moreover, you give BCS result... It does not make sense

table 5. the same... where are the milk yield groups?

Moderate editing of English language

Reviewer 2 Report

The manuscript “Effect of parity, body condition score at calving, and milk yield on the metabolic profile of Gyr cows in the transition period” by Breda et al. analyzes the impact of parity and BCS on milk yield and metabolic profile during transition period in Gyr cows. The topic is of interest and the use of metabolic profile in relatively high-producing Gyr cows is novel. However, there are flaws/confusion in the experimental design that need to be addressed. Moreover, the interpretation of results should be improved. The lack of line numbers made me difficult to put specific comments in the right place, I hope the authors will understand.

Simple summary: greater tolerance than what? Less predisposed than what? Moreover, is it really yield comparable to taurine dairy breeds? Maybe it is true for Jerseys, but not at all for Holsteins (as the authors rightly stated in the discussion section).

Abstract: “…the effects of parity, BCS at calving, or milk yielded.” Please change milk yielded to yield.

As you measure most of the metabolites in serum but glucose in plasma, I would suggest using blood metabolites when referring broadly to the metabolic profile

Introduction: All cattle produce milk in pasture systems, please rephrase

“The importance of the Gyr breed in Brazilian dairy farming is due, in addition to milk production, to its use in crosses with the Holstein breed, which produces Girolando cattle (Holstein × Gyr) that are extremely productive and adapted to the environmental condi-tions typical of tropical or subtropical climates [5].” It is not relevant to the topic of the manuscript. Consider removing?

Materials and Methods:

Why did you include cows in the first months of the rainy season (November and December)? The differences in temperature and humidity can affect the results of the study. Did you record THI?

Based on study duration and herd size, it would appear that all the cows in the herd calving in the study period were included in the study. Is it true? In the text, you said that “cows were selected”. If so, how?

The description of the grouping strategy might appear confusing to the reader. Please rephrase and provide parity distribution in the BCS and yield groups. How did you select NBCS cows among biparous and multiparous? Randomly?

“…most of the cows selected for the study, except the primiparous cows, were redistributed into two other groups according to milk yield…” How did you choose these cows? Why did you exclude 7 cows?

The average yield in the first 60 dim? If you measured milk yield every 15 days, is the average of the first 4 measures? It is not clear how, when, and for how long you measured milk yield

“Cows with dystocia or twin pregnancies were excluded…” Were they included in the 85 cows or they were excluded before?

Please provide starch concentration in the diets. What is lactation concentrate? Are you sure of the kg of DM of the prepartum diet?

[20], [21] and throughout. Please provide the first author

How did you obtain plasma? Blood samples collected for plasma should be refrigerated before centrifugation, in particular to determine glucose

High lipomobilization is not really a metabolic imbalance. References you cited just provide thresholds for association with increased risk for other subsequent diseases. NEFA or BHB higher than the thresholds defined are common in healthy postpartum cows that produce much more milk than those in this study.

Standard deviation should be replaced by standard error.

Results:

Table 2: Please explain better what the authors mean by total dim. If the means are not different, there is no need to use superscripts (e.g. for BCS groups)

Tables 3, 4, and 5 are not useful to the readers in their present form. For each parameter, you should provide a line with the means of each parity, BCS, or yield, as you did in the supplementary tables. I would suggest putting in the main text the supplementary tables instead of these. Figures might also be useful to visualize results, considering the drastic changes happening around calving.

Discussion:

Overall, the discussion section is a series of paragraphs for each parameter analyzed. A more organic discussion would be easier to follow.

The reference (Angelo et al., 2022) should be formatted properly.

“Values such as 6 to 7 kg/day (Angelo et al., 2022), 12 to 13 kg/day [28], 14 kg/day [29], and 11 to 15 kg/day [30] have been reported in Gyr cows.” I’d suggest changing it to “Values ranging from 6 to 15 kd/d have been…”

Differences in yield among parity are mainly due to body size, maturity, and mammary gland activity.

Multiparous did not lose BCS in early lactation? This is odd. Do you have an explanation for that? The low sample size might have affected this result, or the scoring system used might not have been able to detect differences. Edmonson method was developed for Holstein cows. This aspect should be addressed properly.

The lack of difference in NEFA and BHB between parities despite the different yields might be related to liver function.

“Gyr cows appeared to be more adapted to high NEFA”. This is a speculation and data don’t support this statement. The thresholds used are suggestive of a risk. Similar values are commonly found in healthy but high-producing cows.

“More studies with Gyr and other dairy Zebu cows, with a greater number of observations, are needed to establish the association of certain cutoff points for NEFA and BHB with the occurrence of diseases in the transition period, as the critical values assumed for Holstein cows should probably not be taken as a parameter for Zebu cattle” This is an important sentence. I’d suggest highlighting it. Even among taurine breeds, probably specific reference intervals are needed, so this is even more relevant in Gyr cows (and likely not only for NEFA and BHB).

Round 2

Reviewer 1 Report

minor comments:

- avoid explaining acronyms that you not use in the abstract

- tables must be reviewed. Superscripts are not ok

Minor editing of English language required

Author Response

RESPONSE TO THE EDITOR AND REVIEWERS (round 2)

Manuscript ID: animals-2389041

Title: Effect of parity, body condition score at calving, and milk yield on the metabolic profile of Gyr cows in the transition period

Journal: Animals

Dear editor and reviewers

We are grateful for additional suggestions.

Below are responses to each of the comments.

Sincerely, the authors.

REVIEWER 1

Comments and Suggestions for Authors: minor comments:

Reviewer: avoid explaining acronyms that you not use in the abstract

Authors: Done. We suppress AST and GGT.

Reviewer: tables must be reviewed. Superscripts are not ok

Authors: Sorry. We were not able to identify the error with the superscript letters of the tables. In Table 2, superscript letters were deleted when there was no difference between groups. In Tables 3, 4, and 5, superscript letters represent differences in relation to time. The effect of the factor “day relative to calving” was significant for all variables studied, regardless of how the cows were grouped.

Reviewer 2 Report

The authors have addressed some of my previous comments. However, there are still important aspects that need to be clarified. Materials and methods are not presented in an understandable way. Presentation of results is poor and information is missing (supplementary materials should include only minor or side results not discussed in the main text). I would suggest using figures. I appreciated that the authors tried to solve the issue with line numbers but unfortunately they are still wrong. I referred my comments to the corresponding line and page.

Simple summary: Please change using with rearing.

P2.L7: I would assume you are referring to “some” dairy breeds bred under these conditions. Otherwise, this statement is not correct, please clarify.

L29: Please change to “…and mineral metabolism during the transition period in Gyr cows selected for high milk yield.”

L47-50: 26+21+17=64 and not 85. Please clarify.

L50-P3.L3: Again, number of subjects is wrong. The authors need to properly describe how they selected/enrolled the cows and how they were classified. If there are 17 multiparous, how can the HBCS group composed exclusively of multiparous cows (n=21) and the NBCS group include again multiparous cows (n=17). Now there appears to be 38 multiparous cows. Do you mean biparous and multiparous together?

L8-10: Again, 52 biparous and multiparous cows. Previously they were 38 (21+17). If the authors do not clarify, those numbers will not make sense.

Table 1: Is really the unit kg DM/d (I assume it is kg as fed)? Anyway, is it common to fed Gyr cows with such high levels of energy (particularly prepartum)? “Concentrated lactation ration” is even less clear than before. Please specify the composition of the concentrate.

P4.L9: Blood collected for plasma analyses need to be refrigerated. The authors stated that they did not cooled samples. Thus, glucose values are not reliable.

Table 2: Is the average milk production reported the average of the whole lactation?

Table 3, 4, 5 are useless in their current form. The readers should not need to check the supplementary materials for the main results of the study. The presentation of the results needs to be changed. How can you evaluate the differences between groups without having their means? I’d suggest using figures to present results in a clear and concise way. A possible idea is to show 3 panels (one for parity, one for BCS, and one for yield groups) with the trends of blood metabolites during the transition period and 1/3 tables with the p-values.

Please change “different letters represent differences between moments” with “different letters represent differences between timepoints (or sampling days, or something along those lines)

P8.L4: In P5.L16 the authors stated that BCS varied with parity but here they said the opposite. Then, if BCS did not vary with parity, how can be different at 7 and 42 d? Is this a typo? Moreover, you should discuss more the lack of BCS drop in multiparous. It seems that BCS actually dropped, but less than in the other cows, and, because of the sample size or the BCS method, you were not able to detect this drop.

L20-26 are very confusing. First NEFA and BHB are not related to parity, then they were higher in primiparous, and finally lower. Bad connections between sentences.

Overall, English is fine.

Author Response

RESPONSE TO THE EDITOR AND REVIEWERS (round 2)

Manuscript ID: animals-2389041

Title: Effect of parity, body condition score at calving, and milk yield on the metabolic profile of Gyr cows in the transition period

Journal: Animals

Dear editor and reviewers

We are grateful for additional suggestions.

Below are responses to each of the comments.

Sincerely, the authors.

REVIEWER 2

Comments and Suggestions for Authors

The authors have addressed some of my previous comments. However, there are still important aspects that need to be clarified. Materials and methods are not presented in an understandable way. Presentation of results is poor and information is missing (supplementary materials should include only minor or side results not discussed in the main text). I would suggest using figures. I appreciated that the authors tried to solve the issue with line numbers but unfortunately they are still wrong. I referred my comments to the corresponding line and page.

Reviewer: Simple summary: Please change using with rearing.

Authors: Done.

Reviewer: P2.L7: I would assume you are referring to “some” dairy breeds bred under these conditions. Otherwise, this statement is not correct, please clarify.

Authors: Done.

Reviewer: L29: Please change to “…and mineral metabolism during the transition period in Gyr cows selected for high milk yield.”

Authors: Done.

Reviewer: L47-50: 26+21+17=64 and not 85. Please clarify.

L50-P3.L3: Again, number of subjects is wrong. The authors need to properly describe how they selected/enrolled the cows and how they were classified. If there are 17 multiparous, how can the HBCS group composed exclusively of multiparous cows (n=21) and the NBCS group include again multiparous cows (n=17). Now there appears to be 38 multiparous cows. Do you mean biparous and multiparous together?

L8-10: Again, 52 biparous and multiparous cows. Previously they were 38 (21+17). If the authors do not clarify, those numbers will not make sense.

Authors: 85 cows is the sum of 26 primiparous cows, 21 biparous cows, 17 multiparous cows (all with normal BCS at calving), and 21 HBCS cows. We have made a few changes to the text to make it clearer.

Reviewer: a) Table 1: Is really the unit kg DM/d (I assume it is kg as fed)? b) Anyway, is it common to fed Gyr cows with such high levels of energy (particularly prepartum)? c) “Concentrated lactation ration” is even less clear than before. d) Please specify the composition of the concentrate.

Authors: a) We thank. The reviewer is correct. We corrected the Table. b) No. This was exaggerated on the farm where the study was conducted. c) We corrected. d) Unfortunately we cannot fulfill this request because we do not know the specific composition.

Reviewer: P4.L9: Blood collected for plasma analyses need to be refrigerated. The authors stated that they did not cooled samples. Thus, glucose values are not reliable.

Authors: It is the first time we have heard this criticism. In our studies carried out on more distant farms, the samples are always refrigerated in coolers containing recyclable ice, since centrifugation for plasma separation can occur a few hours after collection due to the time required to travel to the laboratory. In the study presented here, it was not necessary, as sample processing occurred quickly because we were housed on the farm itself. In addition, we used vacuum flasks containing EDTA and sodium fluoride in order to inhibit glycolysis in vitro.

Reviewer:Table 2: Is the average milk production reported the average of the whole lactation?

Authors: Yes.

Reviewer: a) Table 3, 4, 5 are useless in their current form. The readers should not need to check the supplementary materials for the main results of the study. The presentation of the results needs to be changed. How can you evaluate the differences between groups without having their means? I’d suggest using figures to present results in a clear and concise way. A possible idea is to show 3 panels (one for parity, one for BCS, and one for yield groups) with the trends of blood metabolites during the transition period and 1/3 tables with the p-values.

  1. b) Please change “different letters represent differences between moments” with “different letters represent differences between timepoints (or sampling days, or something along those lines).

Authors: a) We respect the reviewer's opinion, but we understand that the presentation of results is adequate and we would like not to modify the tables. The justification arguments for this were already presented in previous responses to the editor and reviewers presented in the first round of evaluation. b) Done.

Reviewer: P8.L4: In P5.L16 the authors stated that BCS varied with parity but here they said the opposite. Then, if BCS did not vary with parity, how can be different at 7 and 42 d? Is this a typo? Moreover, you should discuss more the lack of BCS drop in multiparous. It seems that BCS actually dropped, but less than in the other cows, and, because of the sample size or the BCS method, you were not able to detect this drop.

Authors: Thanks for the comment. This was wrong and we fixed it.

Reviewer: L20-26 are very confusing. First NEFA and BHB are not related to parity, then they were higher in primiparous, and finally lower. Bad connections between sentences.

Authors: These are conflicting results from the literature on the subject, which may be subject to different influencing factors. The most relevant evidence from the study in Gyr cows is “These findings suggest that under good nutritional conditions, parity does not influence NEFA and BHB concentrations before and after calving”.
